# MRI Liver Imaging Integrated with Texture Analysis in Native Liver Survivor Patients with Biliary Atresia after Kasai Portoenterostomy: Correlation with Medical Outcome after Surgical Treatment

**DOI:** 10.3390/bioengineering10030306

**Published:** 2023-02-28

**Authors:** Martina Caruso, Arnaldo Stanzione, Carlo Ricciardi, Fabiola Di Dato, Noemi Pisani, Gregorio Delli Paoli, Marco De Giorgi, Raffaele Liuzzi, Carmine Mollica, Valeria Romeo, Raffaele Iorio, Mario Cesarelli, Arturo Brunetti, Simone Maurea

**Affiliations:** 1Department of Advanced Biomedical Sciences, University of Naples “Federico II”, 80131 Naples, Italy; 2Department of Electrical Engineering and Information Technology, University of Naples “Federico II”, 80125 Naples, Italy; 3Bioengineering Unit, Institute of Care and Scientific Research Maugeri, 82037 Telese Terme, Italy; 4Department of Translational Medical Sciences, University of Naples “Federico II”, 80131 Naples, Italy; 5Institute of Bio-Structures and Bio-Imaging of the National Research Council (CNR), Via Tommaso De Amicis, 80145 Naples, Italy

**Keywords:** biliary atresia, liver tissue, MRI, quantitative imaging, texture analysis

## Abstract

Kasai portoenterostomy (KP) plays a crucial role in the treatment of biliary atresia (BA). The aim is to correlate MRI quantitative findings of native liver survivor BA patients after KP with a medical outcome. Thirty patients were classified as having ideal medical outcomes (Group 1; *n* = 11) if laboratory parameter values were in the normal range and there was no evidence of chronic liver disease complications; otherwise, they were classified as having nonideal medical outcomes (Group 2; *n* = 19). Liver and spleen volumes, portal vein diameter, liver mean, and maximum and minimum ADC values were measured; similarly, ADC and T2-weighted textural parameters were obtained using ROI analysis. The liver volume was significantly (*p* = 0.007) lower in Group 2 than in Group 1 (954.88 ± 218.31 cm^3^ vs. 1140.94 ± 134.62 cm^3^); conversely, the spleen volume was significantly (*p* < 0.001) higher (555.49 ± 263.92 cm^3^ vs. 231.83 ± 70.97 cm^3^). No differences were found in the portal vein diameter, liver ADC values, or ADC and T2-weighted textural parameters. In conclusion, significant quantitative morpho-volumetric liver and spleen abnormalities occurred in BA patients with nonideal medical outcomes after KP, but no significant microstructural liver abnormalities detectable by ADC values and ADC and T2-weighted textural parameters were found between the groups.

## 1. Introduction

Among the disorders causing neonatal cholestasis, liver biliary atresia (BA) requires surgical treatment for biliary drainage and represents one of the main indications for liver transplantation in children [1]. Indeed, progressive liver cirrhosis may occur in these young patients if not properly treated, and the cornerstone of surgical management is the procedure known as Kasai portoenterostomy (KP) [2]. For those who undergo KP, a diagnostic follow-up relying on clinical and laboratory tests, as well as imaging studies, is usually scheduled to monitor the hepatic morpho-functional changes that may occur [3,4]. Regarding imaging modalities, ultrasound (US) is regarded as a valuable noninvasive diagnostic approach, despite the recognized limitations linked to its subjective qualitative nature [5]. On the other hand, an objective assessment of the liver parenchyma would be desirable to reliably and timely detect early tissue changes suggestive of chronic liver disease (CLD); thus, quantitative imaging modalities have been proposed. Among them, US shear-wave elastography, which allows for the evaluation of liver stiffness, has been found to be useful in identifying high-risk patients with CLD for liver transplantation [6,7]. In the setting of CLD in BA patients after KP, the role of magnetic resonance imaging (MRI) for the identification of liver abnormalities and/or for prognostic assessment has also been explored, and this has provided morphological information thanks to T1- and T2-weighted sequences [8,9]. More recently, an imaging parameter obtained from a MRI diffusion-weighted sequence (DWI), known as the apparent diffusion coefficient (ADC), was proposed as a quantitative biomarker to assess the severity of CLD [10,11]. ADC values represent a quantitative measure of water-molecule diffusion and are expressed as 10^−3^ mm^2^/s. ADC is inversely related to tissue cellularity and is strongly affected by molecular viscosity, the permeability of the membrane separating the intra- and extracellular compartments, and active transport and flow. Interestingly, the search for imaging biomarkers has increased in recent years thanks to the emergence of radiomics, a complex multistep process that allows for the extraction of a large number of computational quantitative features from digital medical images. These parameters represent a measure of image heterogeneity at the pixel level, which reflect underlying tissue changes at the microscopic level. Radiomics features can be used alone or integrated with clinical features to build predictive models and decision-support tools that could aid physicians in clinical practice. Texture analysis is one of the approaches used in radiomics to obtain quantitative parameters related to the spatial distribution of pixel intensity levels within an image [12].

Previously, a good correlation was reported between US and/or MRI qualitative imaging findings and the medical outcome of BA patients after KP; thus, a potential role of US and MRI findings to predict the long-term medical outcome in such patients has been suggested [13,14]. However, to the best of our knowledge, ADC and texture parameters were not previously investigated in such patients. Thus, the purpose of the present study is to correlate liver quantitative MRI findings integrated with texture analysis with the medical outcome in native liver survivor BA patients after KP.

## 2. Materials and Methods

### 2.1. Patient Population

Patients with BA and surviving with a native liver after KP were retrospectively evaluated in our institution from September 2016 to January 2021. The exclusion criteria consisted of prior liver transplantation or planned liver transplantation, and imaging follow-up <5 years after KP. The patients were classified as having ideal or nonideal medical outcomes after KP according to the proposed clinical criteria (Table 1) [3,4].

In detail, patients with ideal medical outcomes show normal values of all laboratory parameters mentioned in Table 1 and no evidence of CLD medical complications, while patients with nonideal medical outcomes show abnormal values of at least one laboratory parameter and/or one CLD complication. Furthermore, the liver disease status was calculated using the Paediatric End-stage Liver Disease (PELD) score for patients younger than 12 years old, and the Model for End-stage Liver Disease (MELD) score for patients older than 12 years old [15]. These scores were developed in the early 2000s to rank patients awaiting liver transplants according to their probability of survival [15]. In detail, the PELD score considers five variables: age, total bilirubin, albumin, INR, and history of growth failure. It was calculated according to the following formula detailed on the homepage of the United Network for Organ Sharing (UNOS): 10 × (0.480 × ln(bilirubin) + 1.857 × ln(INR) − 0.687 × ln(albumin) + 0.436 (if patient under 12 months) + 0.667 (if history of growth failure positive). The minimum score value is −11 [16]. The MELD score considers three variables: creatinine, total bilirubin, and INR. It was calculated according to the following formula: 10 × (0.957 × ln(creatinine) + 0.378 × ln(bilirubin) + 1.120 × ln(INR) + 0.643). The minimum score value is 0 [15].

### 2.2. Magnetic Resonance Imaging

Imaging studies were performed on a 1.5 Tesla scanner (Gyroscan Intera, Philips, Eindhoven, The Netherlands) following the protocol reported in Table 2. All patients were studied in a fasting state with a dedicated phased array, 4-channel surface body coil.

### 2.3. Quantitative Imaging Analysis

For MRI quantitative analysis, liver and spleen volumes were measured by an abdominal radiologist with 15 years of experience in hepatobiliary imaging using a semiautomatic method with OsiriX^®^ version 3.3 software (Geneva, Switzerland) [17,18]. In detail, liver and spleen contours were manually traced at different levels on T2-weighted images with the closed polygon selection tool; then, the remaining boundaries were automatically outlined using the Grow Region (2D/3D Segmentation) tool. The automatically generated outlines were then hand-adjusted. After selecting all of the ROIs within the series, the software automatically calculated the volume by multiplying the surface and slice thickness and then adding up individual slice volumes. Furthermore, the anteroposterior portal vein diameter was measured in mm with calipers at the liver hilum on an axial T2-weighted sequence. For MRI DWI analysis, the mean, maximum, and minimum signal intensities corresponding to ADC values were measured using ROI analysis, as previously described [10]. To obtain a representative ADC value of the liver parenchyma (liver ADC), a radiology resident drew two circular ROIs with a maximal diameter of 1 cm for the right and left liver lobes. ROIs were positioned within 1 cm of the bifurcation of the main portal vein on each side (Figure 1). Motion artefacts, flow artefacts, or signals from the great vessels were not included in the drawn ROIs. The ADC value was calculated according to the equation S = S_0_ × e^−b×ADC^, where S is the signal intensity after application of the diffusion gradient, b is the diffusion factor, and S0 is the signal intensity at b = 0 s/mm^2^. A diffusion factor of b = 800 was used. The liver ADC value was then expressed as a mean value obtained from the measurements on the left and right liver lobes.

For texture analysis, two fixed-diameter circular 2D ROIs were drawn on T2-weighted and ADC axial images by a radiology resident, one in the left lobe and one in the right lobe. Care was taken to avoid vessels and artefacts, similar to what was previously carried out for ADC measurements. Image annotation was performed using dedicated segmentation software (ITK-Snap 3.8.0, Harrisburg, PA, USA) [19]. An example of delineated ROIs for texture analysis can be found in Figure 2. For the extraction of textural features, the software SERA, which is compliant with the Image Biomarker Standardization Initiative, was adopted [20]. From each ROI, 25 gray-level cooccurrence matrix (GLCM)-based features were obtained. For each patient, the mean of the values from the left and right lobe ROI was calculated and used for subsequent analysis. The complete list of calculated features may be found in the Appendix A.

### 2.4. Statistical Analysis

Continuous data are expressed as the median and range or as the mean and standard deviation. Student’s t-test or Mann–Whitney U tests were used to evaluate differences between the groups in continuous variables, and a chi-square test was performed to compare categorical variables. A two-sided *p*-value < 0.05 was considered statistically significant. All statistical analyses were performed with dedicated software (IBM SPSS Statistics, Ver. 26.0, Armonk, NY, USA; Medcalc Software, ver. 19.1, Oostend, Belgium).

## 3. Results

### 3.1. Patient Population

A total of 40 patients were enrolled, but five patients were excluded due to a follow-up timing of <5 years after KP, while the remaining five were patients awaiting a liver transplant. Hence, the final study population consisted of 30 patients (19 male, 11 female; median age = 14.73 years, range 8–29 years), of whom 11 patients had an ideal medical outcome (Group 1), while the remaining 19 patients had a nonideal medical outcome (Group 2) after KP. The point in time when the MRI exams were performed was 15.2 years (range = 10–18 years) after KP for the patients in Group 1 and 15.5 years (range = 8–29 years) after KP for the patients in Group 2.

Table 3 and Table 4 show the clinical and laboratory findings for the patients in Groups 1 and 2, respectively. No differences in sex, age, or height were observed between the two groups (*p* > 0.05 for all). A significant (*p* < 0.001) difference in the MELD score values was observed between the patients in the two groups. The MELD score was significantly higher in the patients in Group 2 (median value = 9) than in the patients in Group 1 (median value = 7). Conversely, no significant difference was found in the PELD score values between the patients in the two groups.

### 3.2. Magnetic Resonance Imaging

The results of the MRI quantitative analysis are reported in Table 5. In particular, a significant difference in the liver and spleen volumes was observed between the patients in Group 1 and those in Group 2. The liver volume was significantly lower in the patients in Group 2 than in the patients in Group 1; conversely, the spleen volume was significantly higher in the patients in Group 2 than in the patients in Group 1 (Figure 3, Figure 4 and Figure 5). However, no difference was found in the portal vein diameter between the two groups. Furthermore, no differences were observed in the liver ADC mean, maximum, and minimum values between the two groups. Similarly, there were no statistically significant differences among the T2-weighted and ADC textural parameters calculated in the analysis; the corresponding results can be found in the Appendix A.

## 4. Discussion

In patients with BA who have undergone KP, the diagnostic evaluation of liver parenchyma structure after surgical treatment is fundamental in assessing patient outcomes to diagnose early liver cirrhosis [2]. For this reason, the utilization of diagnostic imaging parameters to aid in the early detection of liver abnormalities is clinically desirable. In this study, we investigated the correlation between liver quantitative MRI parameters integrated with texture image evaluation and the medical outcome in native liver survivor BA patients after KP [3,4]. In our experiment, we observed a significant difference only in the liver and spleen volumes between the patients in the two groups, and the results showed a reduction in the liver volume associated with an increase in the spleen volume in the patients with nonideal medical outcomes after surgery. On the other hand, no differences were observed in the quantitative conventional liver ADC values and texture parameters between the two groups; thus, no correlation between the MRI imaging characteristics of liver parenchyma structure and the medical outcome of such patients was detected. Thus, although significant morpho-volumetric liver and spleen abnormalities occurred in patients with nonideal medical outcomes after KP, no significant structural liver abnormalities detectable by conventional DWI sequence and texture analysis, potentially suggestive of early detection of liver cirrhosis, were found in these patients compared to those with ideal outcomes after surgical treatment. Nevertheless, the limited sample size should be considered when assessing the reliability of negative findings. Interestingly, the occurrence of the reduced liver volume associated with splenomegaly in patients with nonideal medical outcomes after KP is concordant with the higher value of the MELD score, as well as with the presence of CLD clinical complications in such patients, and this was mainly represented by portal hypertension. However, the lack of difference in the PELD scores between the two groups could be explained by the limited data. Of note, in previous experiences [13,14], a significant difference was similarly observed in spleen size between patients with ideal and those with nonideal medical outcomes after KP, while no difference was found in liver size, even though significant differences occurred in liver surface and morphology, which were qualitatively assessed. The quantitative tridimensional method used in the present study to evaluate liver and spleen volumes, as well as the different patient populations, could explain this discrepancy.

Advanced MRI using the DWI sequence has been proposed as a quantitative marker of the severity of CLD in patients with BA [10,11]. In particular, DWI sequences can be used to assess the diffusion trend of water within biologic tissues by measuring the ADC value determined by the combined effects of capillary perfusion and diffusion [21]. Previously, Mo et al. demonstrated that the ADC value with a b factor of 500 decreased, while biliary cirrhosis, indicated by the PELD, Child-Turcotte (CT), or Child-Pugh (CP) scoring systems, progressed in BA patients, suggesting that this parameter may be useful for the follow-up and long-term monitoring of such patients [10]. Similarly, Peng et al. showed that the right liver-to-psoas ADC ratio could serve as an indicator of the progression of biliary cirrhosis in BA patients; in particular, this parameter was negatively correlated with the Mayo risk score for primary biliary cirrhosis, as well as with the histopathological liver fibrosis score (METAVIR) [11]. These previous findings represented the rationale to test the DWI sequence with ADC analysis in our patient population with BA. Although liver and spleen morphology abnormalities occurred in patients with nonideal outcomes after KP, such as liver volume reduction and splenomegaly, respectively, no difference in the quantitative conventional liver ADC values was observed compared to patients with ideal outcomes. Thus, no liver parenchyma microstructural abnormalities detectable by DWI/ADC MRI sequence were found in the patients with nonideal outcomes, potentially suggesting that the changes in the diffusion trend of water within liver tissue might not be useful as a marker of early fibrosis, even though the small sample size might affect the reliability of our preliminary results.

To further investigate the conditions of liver parenchyma structure in our patient population with BA after KP, we hypothesized that the evaluation of MRI images using texture analysis could be helpful for the early detection of tissue changes related to CLD. The rationale behind texture analysis is that biomedical images contain quantitative diagnostic information that reflects underlying pathophysiological processes, which are detectable with advanced quantitative imaging methods but not with a visual qualitative assessment or a conventional quantitative analysis [12]. For these reasons, we tested MRI DWI textural evaluation in our patient series to assess the conditions of the liver parenchyma structure to detect early potential tissue changes of CLD, expressed by abnormalities in textural parameters. In line with what was found for ADC values, no differences were observed in the results of this advanced imaging analysis between the patients with ideal and those with nonideal medical outcomes after surgical treatment. This result might suggest a comparable liver parenchyma structure between the patients in the two groups, even though laboratory abnormalities, liver volume reduction, and splenomegaly occurred in the patients with nonideal medical outcomes. In addition, we also performed a texture analysis of liver MRI T2-weighted scans since these morphologic images may better reflect the liver parenchyma structure compared to a functional sequence such as DWI; similarly, no differences in textural MRI T2-weighted parameters were found between the patients with ideal and those with nonideal medical outcomes after KP. However, the small sample size of our series should be considered when assessing the reliability of the corresponding results; for instance, Varghese et al. underlined the limitations of imaging texture analysis when a small sample size with less than 50 patients is investigated. Therefore, the lack of significance for texture features in our cohort does not allow us to make definite claims [22].

This study has some limitations. First, the sample size is small and can have significant implications due to underpowered results, even though the low incidence of BA should be emphasized. Second, the retrospective nature of our study should be considered a limitation, requiring future prospective multicentre studies in larger patient groups. Third, the lack of a pathological standard of reference for liver tissue is not negligible. Fourth, while the circular ROI used for texture analysis in the present study can be considered a strategy to obtain virtual samples of liver tissue, it should be acknowledged that this technical approach might be limited in the assessment of those parenchymal changes occurring in diffuse disease.

In conclusion, these preliminary results suggest that although significant quantitative morpho-volumetric liver and spleen abnormalities occurred in BA patients with nonideal medical outcomes after KP, no significant structural liver abnormalities detectable by conventional DWI sequence and texture analysis of ADC map and T2-weighted images, potentially aiding early cirrhosis detection, were found in such patients compared to those with ideal outcomes after surgical treatment. The limited sample size of our patient series significantly affects the reliability of the corresponding texture MRI results; therefore, additional studies are required to widely investigate the potential role of quantitative MRI integrated with texture analysis in this clinical field.

## Figures and Tables

**Figure 1 bioengineering-10-00306-f001:**
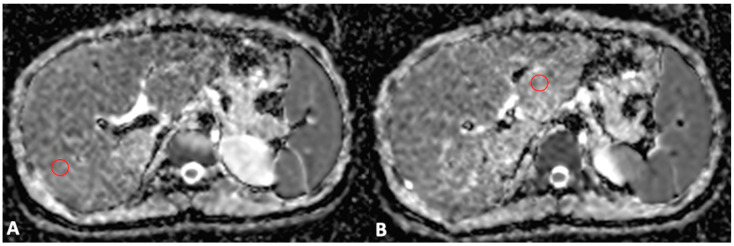
Axial MRI ADC map scans through the right (**A**) and left (**B**) portal veins. Two ROIs with a maximal diameter of 1 cm were drawn for the right (**A**) and left (**B**) lobes, excluding the great vessels and motion artefacts, to obtain a representative ADC value of the liver parenchyma.

**Figure 2 bioengineering-10-00306-f002:**
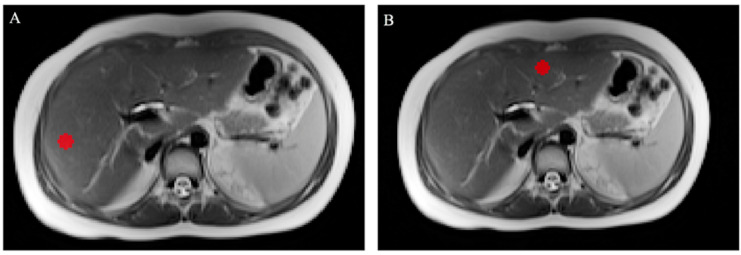
Axial T2-weighted MRI scans showing the right (**A**) and left (**B**) lobe ROIs drawn for texture analysis; care was taken to exclude the great vessels and motion artefacts.

**Figure 3 bioengineering-10-00306-f003:**
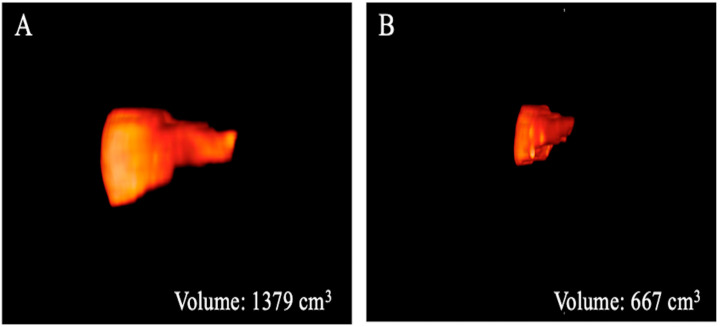
Three-dimensional volume renderings of the whole liver volumes as obtained from the T2-weighted axial sequence segmentation. The example shows a patient from Group 1 (**A**) with a greater volume compared to that from Group 2 (**B**).

**Figure 4 bioengineering-10-00306-f004:**
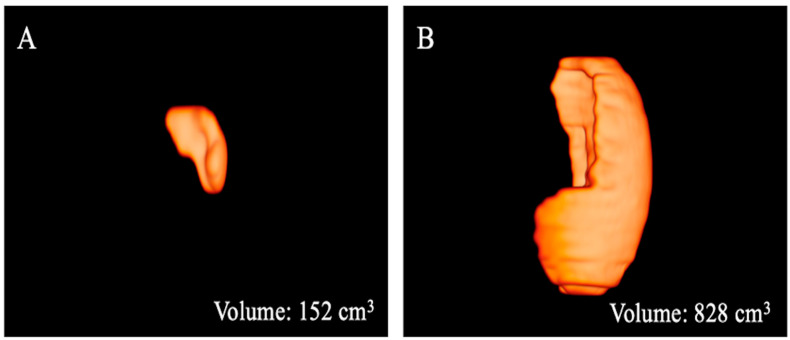
Three-dimensional volume renderings of the whole spleen volumes as obtained from the T2-weighted axial sequence segmentation. The example shows a patient from Group 1 (**A**) with a lower volume compared to that from Group 2 (**B**).

**Figure 5 bioengineering-10-00306-f005:**
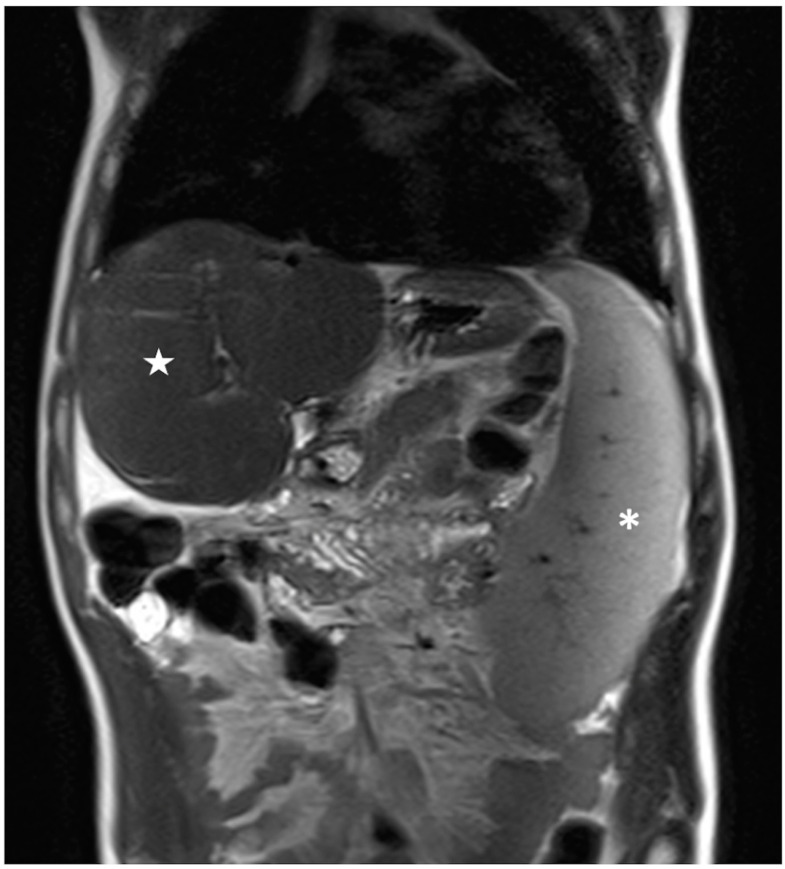
Coronal T2-weighted image of a patient belonging to Group 2 showing large splenomegaly with a total volume of 829 cm^3^ (white asterisk); conversely, the liver shows a reduced volume of 760 cm^3^ (white star).

**Table 1 bioengineering-10-00306-t001:** Laboratory and clinical criteria of medical outcome.

	Classification Criteria of Medical Outcome *	Ideal Medical Outcome	Nonideal Medical Outcome
**Laboratory parameters**	White cell count	≥4000/mm^3^	<4000/mm^3^
	Platelet count	≥150,000/mm^3^	<150,000/mm^3^
	Total bilirubin	≤1.2 mg/dL	>1.2mg/dL
	Albumin	≥3.5 g/dL	<3.5 g/dL
	International Normalized Ratio (INR)	≤1.2	>1.2
	Alanine aminotransferase (ALT)	≤40 UI/l	>40 UI/l
	Aspartate aminotransferase (AST)	≤40 UI/l	>40 UI/l
	γ-glutamyl transpeptidase (GGT)	≤55 UI/l	>55 UI/l
**Complications of CLD**	Portal hypertension	Absent	Present
	Variceal bleeding	Absent	Present
	Fractures	Absent	Present
	Hepatopulmonary syndrome	Absent	Present
	Porto-pulmonary hypertension	Absent	Present
	**Cholangitis in the 12 months preceding imaging exams**	Absent	Present

* The medical status was established according to the criteria of Ng et al. [4] and Lee et al. [3].

**Table 2 bioengineering-10-00306-t002:** MRI protocol performed with 1.5-T MR scanner.

	Axial T1 FFE	Axial and Coronal T2 SSTSE	DWI B Values = 50/400/800 s/mm^2^	MR Cholangiopancreatography (MRCP)
Repetition Time (TR)	214 ms	417 ms	5100 ms	1050 ms
Echo Time (TE)	46 ms	80 ms	68 ms	2600 ms
Flip Angle	80°	80°	90°	
Acquisition Matrix	192 × 512	192 × 512	128 × 80	256 × 320 (reconstruction matrix 256 × 512)
Slice thickness	5 mm	5 mm	5 mm	40 mm

FFE = Fast-Field Echo Sequence. SSTSE = Single-Shot Turbo Spin Echo Sequence.

**Table 3 bioengineering-10-00306-t003:** Laboratory and clinical findings of the patients in Group 1.

#	Sex	Age (Years)	Height (cm)	Laboratory Abnormalities °	CLD Complications	PELD ^§^	MELD ^§^
1	M	10	147	-	-	−10	/
2	M	16	188	-	-	/	7
3	F	16	160	-	-	/	7
4	M	14	171	-	-	/	7
5	F	15	165	-	-	/	6
6	M	10	135	-	-	−9	/
7	F	14	169	-	-	/	7
8	M	18	174	-	-	/	8
9	M	18	184	-	-	/	7
10	M	17	174	-	-	/	7
11	F	17	165	-	-	/	6

° Abnormal values out of the normal range. - = not present. § = Freeman2002. PELD = Paediatric End-stage Liver Disease. MELD = Model for End-stage Liver Disease.

**Table 4 bioengineering-10-00306-t004:** Laboratory and clinical findings of the patients in Group 2.

#	Sex	Age (Years)	Height (cm)	Laboratory Abnormalities °	CLD Complications	PELD ^§^	MELD ^§^
1	M	15	166	WBC	Portal hypertension, cholangitis	/	7
2	M	9	137	WBC, PLT	Portal hypertension	−8	/
3	F	11	136	AST, ALT, WBC, PLT	Portal hypertension, cholangitis	−8	/
4	F	15	153	PLT	Portal hypertension, variceal bleeding, cholangitis	/	9
5	M	12	161	ALT, WBC, PLT	Portal hypertension,	−8	/
6	M	29	175	TB, PLT	Portal hypertension,	/	9
7	M	11	149	GGT, WBC, PLT	Portal hypertension,	−7	/
8	F	14	162	AST, WBC, PLT	Portal hypertension, cholangitis	/	9
9	M	18	180	TB	-	/	9
10	M	14	163	TB	-	/	9
11	M	16	167	TB	Cholangitis	/	10
12	M	9	134	PLT	-	−10	/
13	F	17	160	TB	Cholangitis	/	8
14	F	8	128	ALT, WBC	Cholangitis	−10	/
15	M	17	167	WBC, PLT	-	/	7
16	F	17	153	PLT	Portal hypertension, cholangitis	/	9
17	M	13	155	WBC, PLT	Portal hypertension,	/	9
18	F	14	162	AST, ALT, TB, PLT	Portal hypertension, cholangitis	/	12
19	M	18	174	TB	Cholangitis	/	12

° Abnormal values out of the normal range. - = not present. § = Freeman2002. PELD = Paediatric End-stage Liver Disease. MELD = Model for End-stage Liver Disease.

**Table 5 bioengineering-10-00306-t005:** Results of the quantitative MRI parameters of the patients in both groups.

MRI Parameter	Group 1 (Mean ± SD)	Group 2 (Mean ± SD)	*p* Value
**Liver volume** (cm^3^)	1140.94 ± 134.62	954.88 ± 218.31	**0.02**
**Portal vein diameter** (mm)	9.2 ± 1.6	9.4 ± 1.9	0.70
**Spleen volume** (cm^3^)	231.83 ± 70.97	555.49 ± 263.92	**<0.01**
**Liver ADC mean**	1050.09 ± 114.01	1010.05 ± 84.178	0.33
**Liver ADC minimum**	743.1 ± 134.6	749.7 ± 115.5	0.89
**Liver ADC maximum**	1399.4 ± 158.1	1292.4 ± 113.8	0.07

## Data Availability

Data are not available due to privacy policy.

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
