# Peer review of "MRI Liver Imaging Integrated with Texture Analysis in Native Liver Survivor Patients with Biliary Atresia after Kasai Portoenterostomy: Correlation with Medical Outcome after Surgical Treatment"

_bioengineering, 2023, doi:10.3390/bioengineering10030306_

Round 1

Reviewer 1 Report

This study investigated the correlation between MRI findings and liver texture analysis in biliary atresia patients who underwent Kasai portoenterostomy. For the study a total of 30 patients was ultimately selected (11 with ideal medical outcome and 19 without ideal medical outcome). Both genders were represented in both groups. Age of the patients recruited in the two groups varied from 9 to 29. Well identified Lab parameters and complications of CLD were used to divide the patients between the two groups. The results reported in the study indicate that  MRI detected significant morpho-volumetric liver and spleen abnormalities in patients without ideal medical outcome. However, MRI was not able to detect structural liver abnormalities that were detected by conventional DWI sequence and texture analysis of ADC map, potentially negating the ability of MRI of detecting early cirrhosis. The possibility that the sample size was too small is discussed. 

Author Response

Reviewer 1

This study investigated the correlation between MRI findings and liver texture analysis in biliary atresia patients who underwent Kasai portoenterostomy. For the study a total of 30 patients was ultimately selected (11 with ideal medical outcome and 19 without ideal medical outcome). Both genders were represented in both groups. Age of the patients recruited in the two groups varied from 9 to 29. Well identified Lab parameters and complications of CLD were used to divide the patients between the two groups. The results reported in the study indicate that MRI detected significant morpho-volumetric liver and spleen abnormalities in patients without ideal medical outcome. However, MRI was not able to detect structural liver abnormalities that were detected by conventional DWI sequence and texture analysis of ADC map, potentially negating the ability of MRI of detecting early cirrhosis. The possibility that the sample size was too small is discussed.

We all thank the reviewer for appreciating and reviewing our manuscript.

Reviewer 2 Report

1. Section 2.3. Quantitative Imaging Analysis is not clearly described.

2. The quality of figures are very low. Please enhance the resolution of the figures.

3. Discussion part is not sifficient. You should explain more.

4. Please improve the English written of the manuscript.

5. What is Figure 3? Please clarify it.

Author Response

Reviewer 2

  1. Section 2.3. Quantitative Imaging Analysis is not clearly described.

We added a detailed description of the method used to measure liver and spleen volumes.

  1. The quality of figures are very low. Please enhance the resolution of the figures.

We are sorry that the quality of figures in the manuscript version that you revised is not satisfactory. However, we verified the quality of the original image files and can confirm that all figures have a resolution of 300 dpi.

  1. Discussion part is not sufficient. You should explain more.

In the Discussion section, we put our findings in the context of the relevant literature and tried to formulate hypotheses that could explain our results. During this revision, we made some further amendments. We believe we were now able to cover all the analyses performed in the study. At the same time, we would like to avoid making the Discussion too lengthy and hard to read. However, we would be glad to further expand this section if this was also deemed necessary by the Editors, but in such case, we would kindly ask for some additional details on what could represent a valuable addition (e.g., are there any other relevant articles we should have mentioned? Are there any interesting points that we should have raised? Are there different interpretations of the results that we should present to the readers?).

  1. Please improve the English written of the manuscript.

Thank you for your suggestion. We revised the manuscript under the supervision of a native English speaker and hope to have improved the quality of presentation and writing. However, we also wish to underline that the manuscript has been previously edited for proper English language, grammar, punctuation, spelling, and overall style by one or more of the highly qualified native English-speaking editors at Springer Nature Author Services (verification code 38D3-6222-08D5-3A95-88B1).

  1. What is Figure 3? Please clarify it.

Figure 3 is a three-dimensional volume rendering of the whole liver volumes obtained from the T2-weighted axial sequence segmentation. This is specified in the figure legend.

Reviewer 3 Report

This is a retrospective study evaluating the MRI liver tissue characterization in patients who underwent kasai portoenterostomy for biliary atresia.

1. "ideal" and "nonideal" is difficult to understand. Please explain in the abstract also.

2. Although the title is "quantitative MRI liver tissue characterization", the manuscript only discuss the volume. This point should be modified.

Author Response

Reviewer 3

This is a retrospective study evaluating the MRI liver tissue characterization in patients who underwent kasai portoenterostomy for biliary atresia.

  1. "ideal" and "nonideal" is difficult to understand. Please explain in the abstract also.

We moved up Table 1 and we provided more details regarding the classification criteria of ideal and non-ideal medical outcome both in the abstract and in the main text.

  1. Although the title is "quantitative MRI liver tissue characterization", the manuscript only discusses the volume. This point should be modified.

We appreciate the Reviewer’s suggestion. However, while it is true that volume turned out to be the most promising quantitative parameter in our analysis, several different quantitative parameters were included in our study (e.g., ADC values and textural features). Despite being negative findings, all are reported extensively in the manuscript. Thus, we think that the current title better depicts the whole study design.

Reviewer 4 Report

The authors did a retrospective study of 30 patients, to study the corelation of quantitative MRI Liver tissue characterization integrated with texture analysis in biliary atresia patients surviving with native liver after Kasai portoenterostomy with medical outcome after surgical treatment. The sample size is low, and the results from the study provides little additive information, Moreover, some results are contradictory to current reports. 

The title should be changed, current one is too tedious.

ADC need to be defined.

Author Response

Reviewer 4

The authors did a retrospective study of 30 patients, to study the correlation of quantitative MRI Liver tissue characterization integrated with texture analysis in biliary atresia patients surviving with native liver after Kasai portoenterostomy with medical outcome after surgical treatment. The sample size is low, and the results from the study provides little additive information, Moreover, some results are contradictory to current reports.

The title should be changed, current one is too tedious.

Thank you for your suggestion, we choose this title to accurately present all aspects of our study.

ADC need to be defined.

We added some details about ADC in the introduction.

Round 2

Reviewer 3 Report

The authors have revised the manuscript appropriately. However, I don't think the title "quantitative MRI liver tissue characterization" is not appropriate.

Author Response

Reviewer 3

The authors have revised the manuscript appropriately. However, I don't think the title "quantitative MRI liver tissue characterization" is not appropriate.

Thank you for your suggestion, we propose this new title "MRI Liver imaging integrated with texture analysis in native liver survivor patients with biliary atresia after Kasai portoenterostomy: correlation with medical outcome after surgical treatment."
